# Investigations of a Weathered and Closely Jointed Rock Slope Failure Using Back Analyses

**Kuo-Shih Shao** [1,2], **An-Jui Li** [1,*], **Chee-Nan Chen** [1], **Chen-Hsien Chung** [1], **Ching-Fang Lee** [2] **and Chih-Ping Kuo** [3]

1   Department of Civil and Construction Engineering, National Taiwan University of Science and Technology, Taipei 106335, Taiwan; sks@sinotech.org.tw (K.-S.S.); chee-nan@mail.ntust.edu.tw (C.-N.C.); johnnychung850717@gmail.com (C.-H.C.)
2   Sinotech Engineering Consultants, Inc., Taipei 11494, Taiwan; cflee@sinotech.org.tw
3   Department of Civil Engineering and Environmental Informatics, Minghsin University of Science and Technology, Hsinchu 30401, Taiwan; picnic.kuo@must.edu.tw
*   Correspondence: laj871@mail.ntust.edu.tw

**Abstract:** This study presents the case of a landslide triggered by a high groundwater level caused by several days of continuous rainfall in the northeastern region of Taiwan. The slope where this landslide occurred consists of closely jointed and weathered bedrock. By means of finite element limit analysis and the Hoek–Brown failure criterion, this study performed a slope failure simulation similar to the actual landslide and deduced the reasonable value range for the combination of key Hoek–Brown failure criterion parameters through back analyses. The results indicate that the key parameters affecting the bedrock's slope stability were the geological strength index (GSI) and the disturbance factor (D), whereas the effects of the unconfined compressive strength ($\sigma_{ci}$) were less significant. The results of the back analysis reveal that the suitable D-value range and GSI of closely jointed and weathered sandstone in the northeastern region of Taiwan are 0.8 to 0.9 and 20 to 30, respectively. These back-analyzed value ranges can serve as a reference for broader applications in the preliminary stability analysis of similar rock slopes where it is difficult to perform in situ investigation.

**Keywords:** rock mass disturbance; finite element limit analysis; Hoek–Brown failure criterion; rainfall

## 1. Introduction

Taiwan is an island located on the western side of the Pacific Ocean, along the boundary between the Eurasian Plate and the Philippine Sea Plate. Due to the orogeny that occurs at the convergence of these two tectonic plates, more than 70% of Taiwan's area consists of hillslopes, where the bedrock appears to be more fractured due to crustal stress [1,2]. Therefore, it is not uncommon for engineering projects in Taiwan to encounter hillslopes, and slope stability has become an issue of remarkable importance in geotechnical engineering. Due to its geographical position, Taiwan is subjected to heavy rains and even extremely heavy rains concomitant with typhoons during the summer and the northeastern winds during the winter. Northeastern cloud systems contain moist air masses that form considerable amounts of rainfall over the windward slopes in the northeastern coast of Taiwan, which increases the likelihood of landslides. This situation is exacerbated by extreme precipitation triggered by global warming and climate change. In order to capture more realistic phenomenon of a slope failure, the applied techniques of slope stability analysis methods must constantly be improved. Furthermore, the bedrock structure in Taiwan is often characterized by discontinuities and groundwater-containing fissures that increase the complexities and difficulties faced by geotechnical engineers in their slope stability analysis.

To date, slope stability analysis has consisted of two methods. The first method is the conventional limit equilibrium method (LEM), whereby the driving and resisting forces are used to derive the factor of safety (FS) of the most possible slip plane of a slope. Even

though the LEM has fundamentals of mechanics, the accuracy of the assessment results depends on reasonable and accurate assumptions of a failure plane, and this method is less suitable for slopes with jointed rock masses and discontinuities. The second method is landslide susceptibility analysis (LSA), which is used on wider slope areas and encompasses a broader range of assessments grounded in geological and geographical theories. LSA is performed quantitatively or qualitatively, and both approaches are described as follows.

Qualitative LSA is performed based on landslide topography and geomorphological features [3]. Another method uses weights assigned by experts based on their evaluations of potential landslide factors [4]. Quantitative LSA requires the integration of numerous topographical, geological, regional, and hydrological factors that could trigger a landslide. The landslide susceptibility index is then calculated via multivariate analyses that consist of landslide and nonlandslide weight combinations, followed by the generation of a landslide susceptibility map for a region through a geographic information system (GIS). In quantitative LSA, factor–weight combinations differ by the topographical and geological features of the local region, and multivariate statistical analysis or machine learning model, etc., are often the main analytical methods [5–11]. Both qualitative and quantitative LSA lack thorough assumptions about the mechanical properties of soil or rock mass slopes, and the results are not presented as the FSs with which the geotechnical engineers are familiar. Therefore, quantitative LSA is mostly used as a reference in the preliminary planning stage or route selection stage of civil engineering projects.

Future developments in slope safety assessment techniques will gravitate toward the means to generate analytical results with mechanical and FS implications that can be used for slope stability analysis over large areas and by reasonably taking into account the mechanical properties of the bedrock or soil slopes. Following the enhancements in computing power, two methods—finite element method (FEM) and limit analysis (LA)—have been applied to complex geological conditions [12]. Grounded in the principles of LA, Lyamin and Sloan [13,14] and Krabbenhoft et al. [15] developed the finite element limit analysis (FELA) method as a solution for problems related to slopes that consist of soil, bedrock, or a mixture of both materials. This study examines the factors that affect the failure of weathered and fractured bedrock slopes by performing slope stability analyses using the Hoek–Brown failure criterion [16,17] in conjunction with FELA.

The bedrock structure of Taiwan's hillslopes is more fractured because of crustal stress and strain. Weathering also reduces the strength of the bedrock, while rainwater infiltration further increases the water pressure inside a slope, which heightens the risk of shallow debris type landslides [18,19]. While the Hoek–Brown failure criterion has been used to analyze rock slopes with similar bedrock conditions in other countries [20,21], this criterion has seldom been applied in Taiwan. There is also a scarcity of examinations on localized parameters as well as practical cases for making comparisons. To this end, this study examines the process of the failure of closely jointed weathered bedrock slopes commonly observed in Taiwan. The Hoek–Brown failure criterion was used to examine the applicability of key parameters for rock slope stability analysis, thereby validating the suitability of the Hoek–Brown failure criterion and FELA for analyzing the stability of weathered and fractured rock slopes in Taiwan. The results could be applied in the preliminary engineering analysis of bedrock slopes over large areas in Taiwan.

## 2. Principles of Analysis

### 2.1. Finite Element Limit Analysis (FELA)

FELA, which is a combination of plastic limit theorems and finite element concepts, designates the physical solution of engineering materials under stress within upper and lower bound solutions [13], while approximating ultimate loads that are closer to actual conditions and automatically generates the location where a failure plane has formed. Slope stability analysis was performed in this study by means of the Optum$^{G2}$ software [22]. The theoretical basis is as follows.

For upper bound limit theorem, Figure 1 shows a conceptual model with a kinematically admissible velocity field that satisfies velocity boundary conditions and the flow rule. An element consists of a volume **K** and a surface area **A**. The fixed additional load **f** acts on the volume, while an unknown load **g** acts on the element's corresponding surface area $\mathbf{A_h}$ alongside the velocity boundary conditions **h.** In Equation (1), $\dot{\varepsilon}$ is the plastic strain rate. The objective of minimizing $\mathbf{H^{int}}$ under equal internal and external power dissipation (see Equation (2)) is to obtain a velocity distribution **u** that satisfies deformation compatibility conditions and the plastic flow rule.

$$H^{int} = \int_K \sigma\dot{\varepsilon}dK \tag{1}$$

$$H^{int}_{min} = H^{ext} = \int_{A_f} f^T u dA + \int_{A_g} g^T u dA + \int_K a^T u dK + \int_K b^T u dK \tag{2}$$

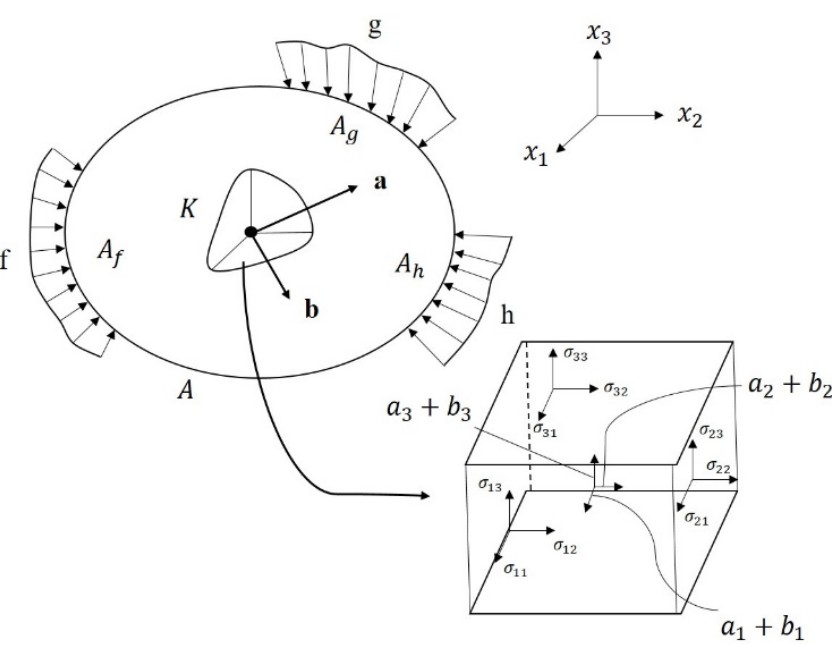

**Figure 1.** Conceptual model of the finite element-based upper bound limit theorem (adapted from Lyamin and Sloan 2002a) [13].

For lower bound limit theorem, a statically admissible stress field must satisfy stress boundary conditions and the yield criterion. A conceptual model of the theorem is shown in Figure 2, in which a geotechnical material is assumed to be a perfectly plastic material that obeys the associated flow rule. An element consists of a volume **K** and a surface area **A**. An unknown traction **g** acts on the element's corresponding surface area $\mathbf{A_g}$, while an unknown traction **f** acts on the corresponding surface area $\mathbf{A_f}$. Additionally, a fixed body force "**b**" and an unknown body force "**a**" act on the entire volume **K**. The finite element-based lower bound limit method derives a stress distribution through Equation (3) that satisfies the entire volume **K** within a statically admissible stress field, balances the fixed traction **f** acting on the surface area $\mathbf{A_f}$, and maximizes the integral **Q**.

$$Q = \int_{A_g} g dA + \int_K a dK \tag{3}$$

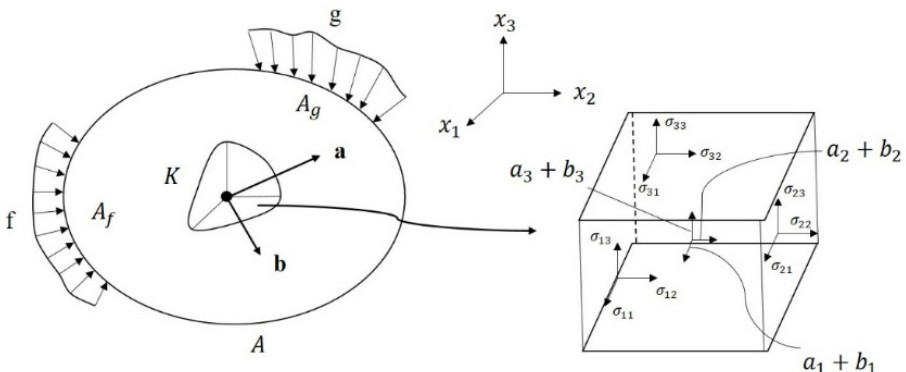

**Figure 2.** Conceptual model of the finite element-based lower bound limit theorem (adapted from Lyamin and Sloan 2002b) [14].

*2.2. Hoek–Brown Failure Criterion*

The Hoek–Brown failure criterion is a nonlinear failure envelope suitable for a rock mass material. Its equation is as follows:

$$\sigma'_1 = \sigma'_3 + \sigma_{ci}\left(m_b\frac{\sigma'_3}{\sigma_{ci}} + s\right)^\alpha \tag{4}$$

where $m_b$, s, and $\alpha$ are rock mass property constants derived, respectively, through the following equations:

$$m_b = m_i \times \exp\left(\frac{GSI - 100}{28 - 14D}\right) \tag{5}$$

$$s = \exp\left(\frac{GSI - 100}{9 - 3D}\right) \tag{6}$$

$$\alpha = \frac{1}{2} + \frac{1}{6}\left(e^{-GSI/15} - e^{-20/3}\right) \tag{7}$$

$\sigma_{ci}$ is the unaxial compressive stress of an intact rock; $\sigma'_1$ and $\sigma'_3$ represent, respectively, the maximum and minimum principal stresses subjected onto the rock mass; $m_i$ is the intact rock constant; and GSI is the geological strength index that is determined based on the in situ examinations. D is the disturbance factor that represents the degree of disturbance acting on the rock mass.

## 3. Description and Numerical Analysis Model of the Case Landslide

*3.1. Description of the Case Landslide*

Between late November and early December 2020 (27 November to 4 December 2020), the Ruifang District located in the northeastern part of New Taipei City experienced continuous rainfall triggered by wintry northeastern winds. As shown in Figure 3, the district recorded an accumulated rainfall of 600 mm within a week. Subsequently, a landslide occurred at the 12 k + 233 m section of the Taiwan Railways Administration's (TRA) Yilan Line. The location of the landslide is shown in Figure 4a, and the aerial view is shown in Figure 4b. Below the landslide area is the Keelung River, which runs into Taipei City. Fortunately, the landslide did not cause any casualties along the railway. Nonetheless, the two tracks along sections 12 k + 218 m to 255 m of the Yilan Line were buried under 2500 m³ of rock and soil, disrupting all railway services from both directions. When the landslide occurred in the morning of 4 December 2020, the area had recorded a 24-h accumulation of rainfall of 81.8 mm. The accumulated rainfall observation is shown in Figure 5, which indicates that the 24-h accumulation of rainfall in the area had reached the Central Weather Bureau's (CWB) "heavy rain" standard (more than 80 mm/24 h).

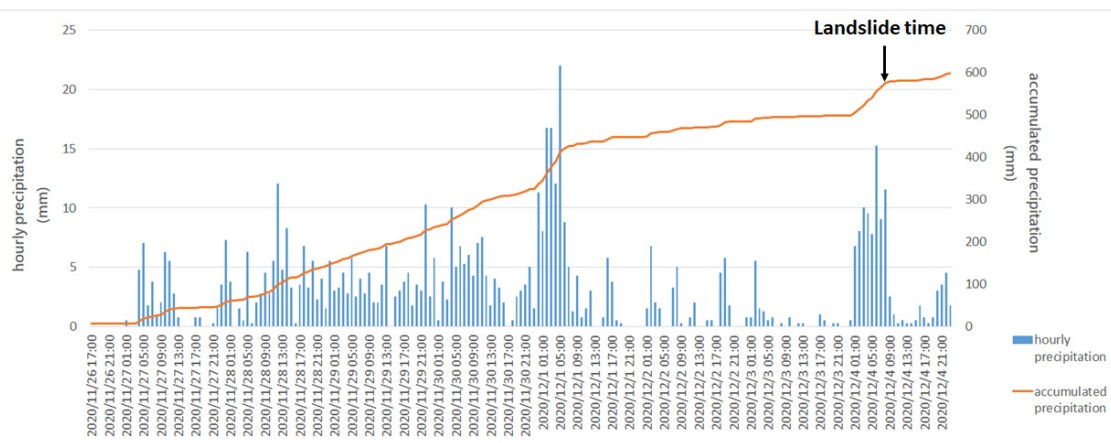

**Figure 3.** Time series plot of rainfall (data acquired from the website "CWB Observation Data Inquire System-CODiS" (available online: https://e-service.cwb.gov.tw/HistoryDataQuery/index.jsp, (accessed on 10 December 2020)) and provided by Ching-Fang Lee/Sinotech Engineering Consultants.).

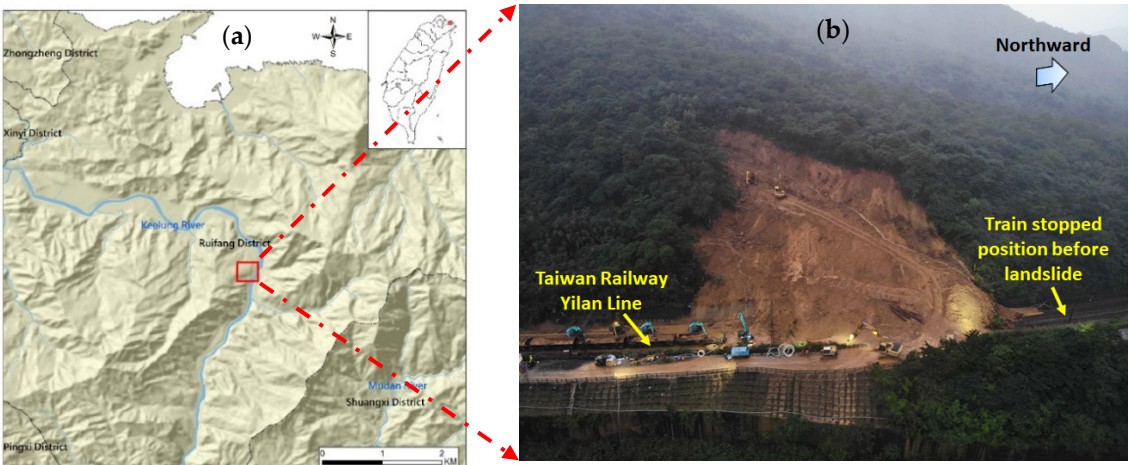

**Figure 4.** Landslide location and its aerial photography. (**a**) Framed area indicating the location of the landslide. (**b**) Aerial photography of emergency repair stage after landslide (courtesy of Sinotech Engineering Consultants, Inc.).

Figure 3 shows that the daily accumulation of rainfall in the landslide area was 300 mm from 27 to 30 November 2020, and this figure rose to 400 mm on 1 December 2020. The rainfall became gradual afterwards, reaching 600 mm during the morning of 4 December 2020, when the landslide occurred. Based on the rainfall accumulated from late November to early December 2020, it is posited that the groundwater level in the area had risen due to high water infiltration into the bedrock slope.

The following depiction of the landslide event is based on multiple news reports at the time. Due to several days of continuous rainfall, the TRA, on 30 November 2020, noticed that debris had fallen at the toe of the slope where the landslide occurred. The section was closely monitored following this incident. At 7:00 a.m. on the day of the landslide (4 December), the driver of a train passing by the section noticed slight fissures and bulging of the shotcrete on the toe along with mudflow. This prompted the TRA to issue a safety warning for the area. At 9:00 a.m., TRA inspection personnel observed that the unstable region of the slope had widened, and they immediately notified the driver of an oncoming train, which came to a halt 30 m away from the site of the landslide. In short, the landslide occurred within 2 h after 7:00 a.m. when the fissures and bulging of the shotcrete were first noticed.

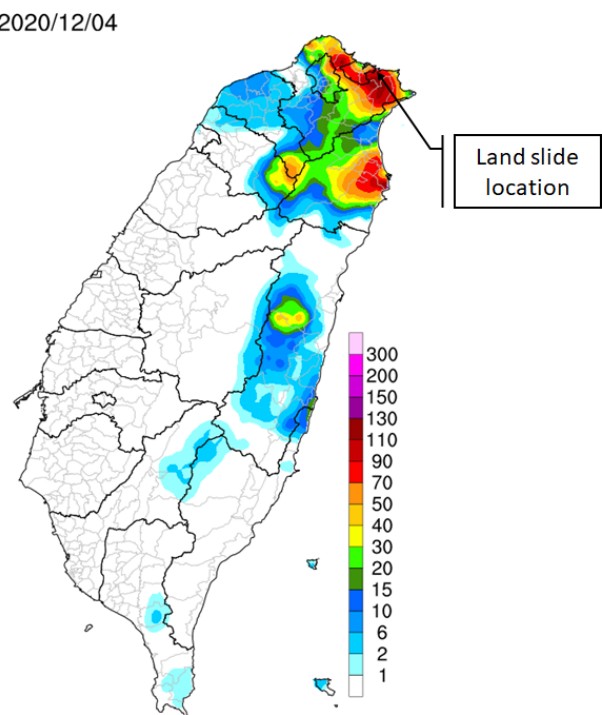

**Figure 5.** Location of the landslide and the level of accumulated rainfall at the time (author edited the original picture from the website of "NCDR Weather and Climate Monitoring" (available online: https://watch.ncdr.nat.gov.tw/watch_history_rainmap (accessed on 10 August 2021))).

Based on the information above, it is posited that the bedrock near the toe of the slope had already failed partially on 30 November. The accumulated rainfall for that day was 300 mm. The active earth pressure, in combination with several days of continuous rainfall, had increased the groundwater pressure, which, in turn, compressed the shotcrete at the toe of the slope to the point that it bulged and fissured, with loose soil flowing out along the process. Meanwhile, the failure plane within the bedrock gradually moved upwards and ultimately triggered the landslide on 9:00 a.m., December 4, at a time when the accumulated rainfall had reached 600 mm.

*3.2. Topography and Geology of the Landslide Area*

Figure 6 shows the pre-landslide topography of the area illustrated by using a digital elevation model (DEM). The average gradient of the 35-meter-tall pre-landslide slope was about 30°. Figure 7a is a regional geological map showing the location of the landslide. The stratum here is part of the Taliao Formation formed during the Miocene epoch. The top part of the stratum primarily consists of calcareous sandstone, while the bottom part primarily consists of gray-black shale. The sandstone occasionally includes shale and siltstone [23]. The primary geological structure of this area is the Houtong Anticline, and the bedrock appears to be more fractured because of bending moments acting on the anticline axial strata near the slope. Figure 7b is a regional rock mass strength map produced by the Central Geological Survey (CGS) and based on Franklin's classification system shown in Figure 7c [24]. By comparing the location of the landslide with the rock mass strength map, the rock mass structure type of the landslide area is classified as blocky fractured and thick layered, with a rock mass discontinuity spacing ranging from 0.2 m to 0.6 m. This differs marginally from in situ inspections. The uniaxial compressive strength of the rock mass ranges from 10 MPa to 25 MPa. By combining the assessment results of the rock mass regional strength and bedrock intactness, the area is classified according to the CGS rock mass strength classification system (Figure 7b) as Type IV, which has moderate but close to poorer strength rock mass.

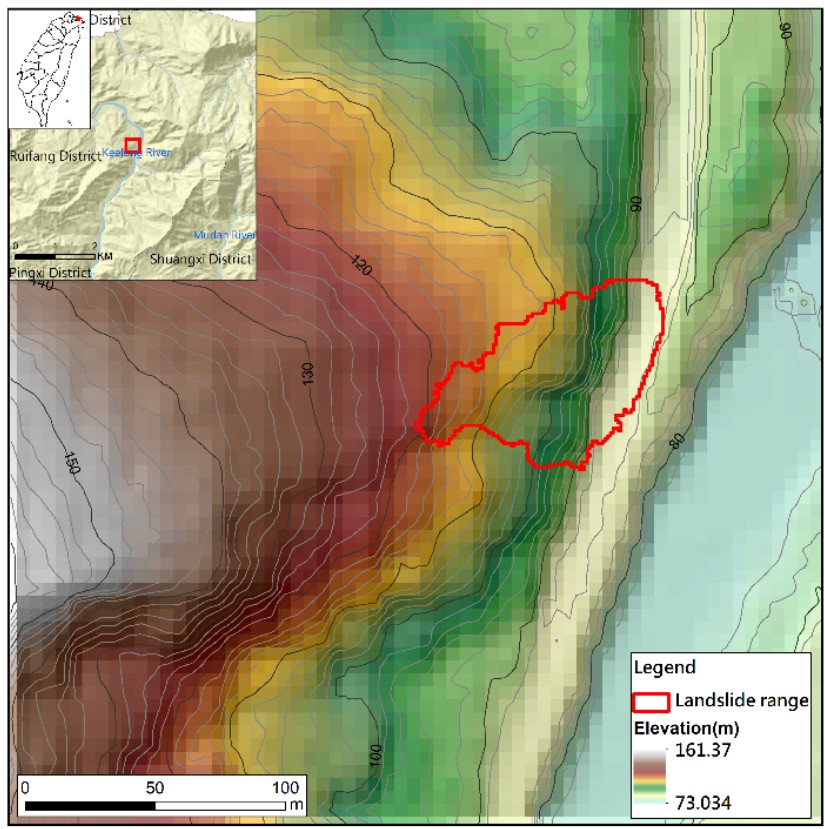

**Figure 6.** Pre-landslide topography and elevation of the area.

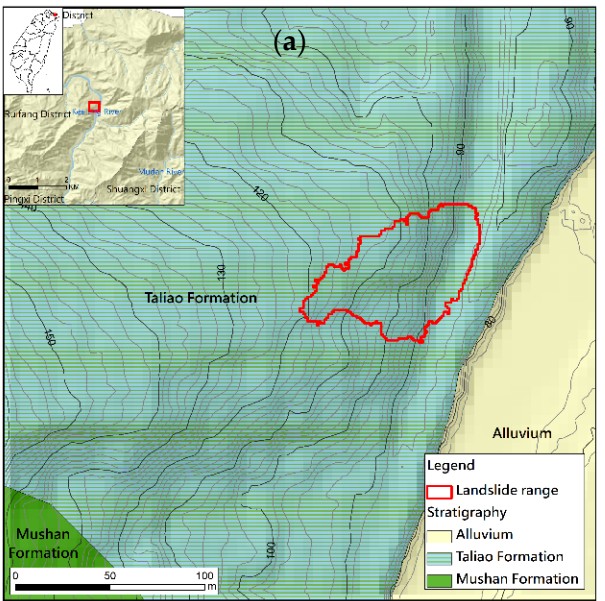

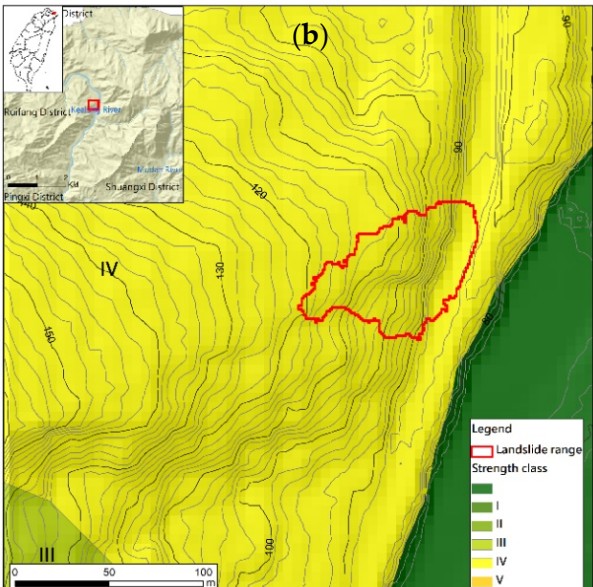

**Figure 7.** *Cont*.

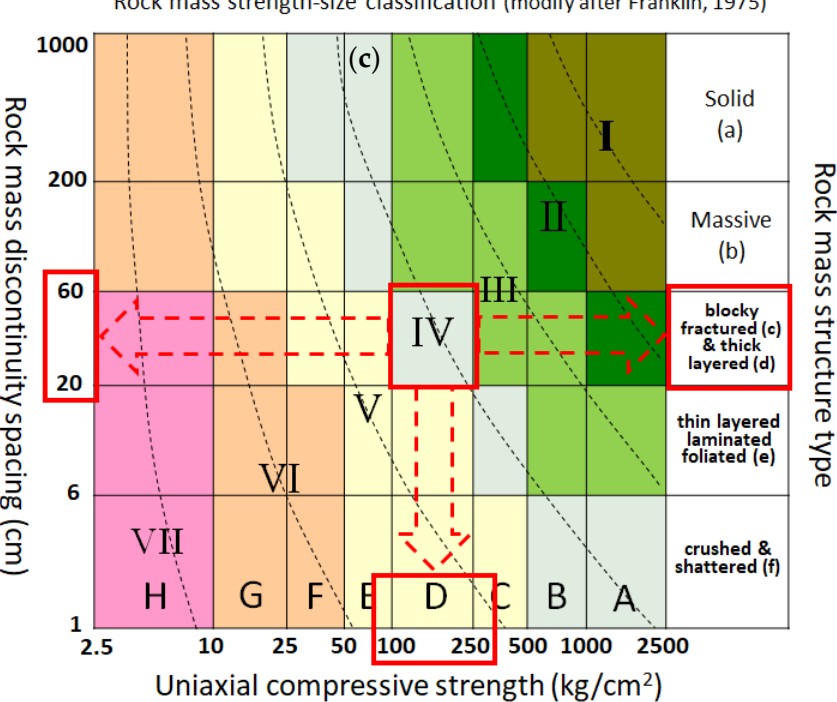

**Figure 7.** Geological map of the study area and bedrock rock mass classification. (**a**) Regional geological map. (**b**) Regional rock mass strength map. (**c**) Classification of rock mass strength by the CGS based on Franklin's classification system [24].

In situ investigation performed by Sinotech Engineering Consultants, Inc. showed that the primary stratum of the slope consists of green-grey sandstone within the Taliao Formation, as well as moderately to highly weathered sandstone blocks located 6 m to 8 m under the slope surface. In addition to stratum plane, the internal structure of the sandstone was cut up by at least four joint sets into smaller fractured blocks, as shown in Figure 8a. As a result of well-developed joints, the weathered and fractured sandstone here increased the groundwater level inside the slope (see Figure 8b). The topographical and geological conditions here served as a reference for devising a numerical slope model for analysis.

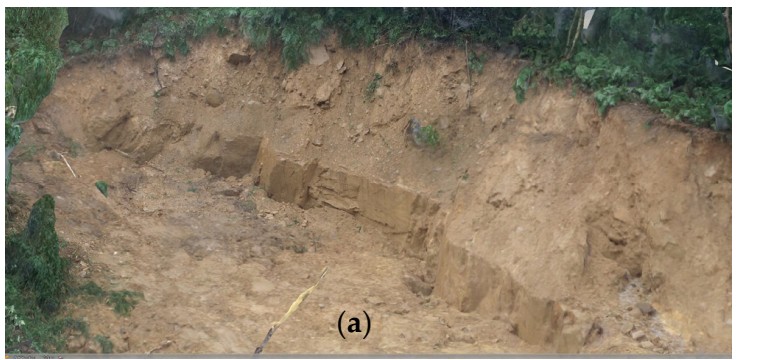
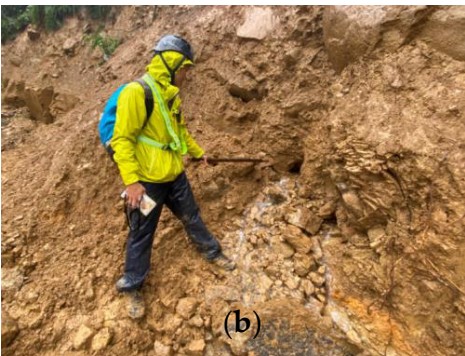

**Figure 8.** Weathered and fractured blocky sandstone and the groundwater situation at the landslide area (courtesy of Sinotech Engineering Consultants, Inc.). (**a**) Weathered and fractured rock mass at the top section of the landslide. (**b**) Percolation of groundwater near the surface of the slope.

### 3.3. Development of a Finite Element Limit Analysis (FELA) Model and Its Parameters for the Slope

Figure 9 shows the topographic profile of the main landslide section. Overlaying the pre-event topography (DEM, pure ground surface) with the post event topography (DSM,

containing pure ground surface and landforms), the topography reveals that the top part of the slope was the sliding section, and the bottom part was the pileup section. These conditions serve as a topographic reference for the slope model for numerical analysis and for setting the boundary conditions. The slope analysis model generated using Optum$^{G2}$ is presented in Figure 10 alongside the pre-event topography shown in Figure 9b. The model is set to be subjected to plane strain conditions, with the Hoek–Brown failure criterion serving as the failure criterion for the rock mass material of the slope. The following is a list of required parameters for numerical analysis: rock unconfined compressive strength ($\sigma_{ci}$), intact rock constant ($m_i$), rock unit weight ($\gamma$), geological strength index (GSI), and disturbance factor (D). The values of each parameter are listed in Table 1, and the reasons for selecting these parameters are described below.

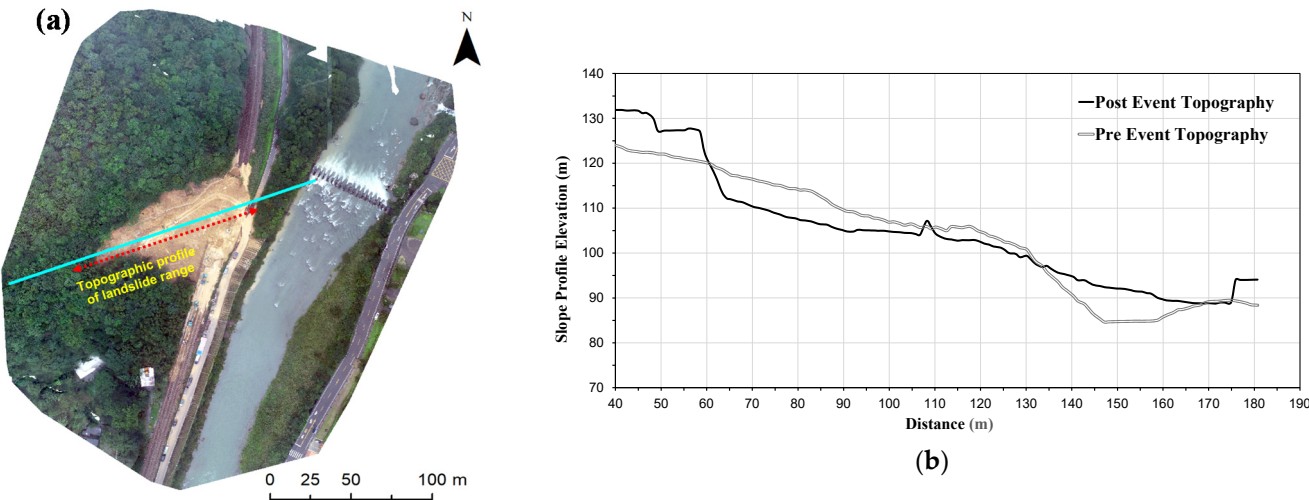

**Figure 9.** Landslide aerial view and its topographic profile. (**a**) Aerial view of the landslide area. (**b**) Topographic profile of the landslide slope (dark line represents the post event topography, light line represents the pre-event topography).

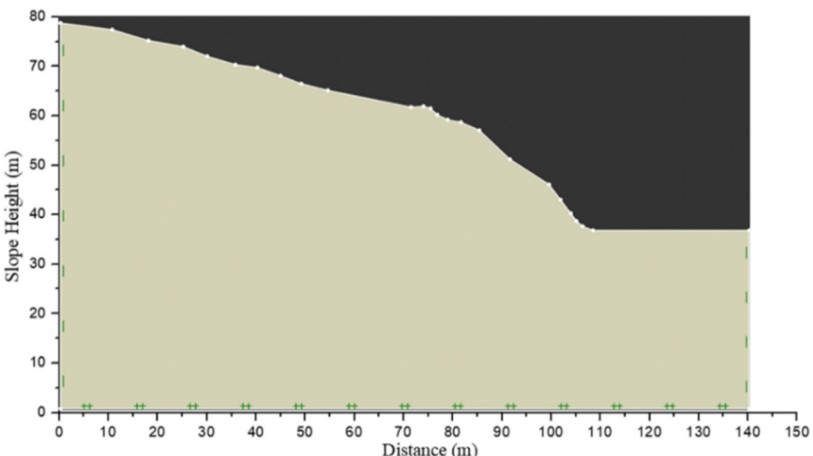

**Figure 10.** Optum$^{G2}$ slope stability analysis model.

**Table 1.** Material parameters for the Hoek–Brown failure criterion used for slope stability analysis in Optum$^{G2}$.

| Parameter | GSI | $\sigma_{ci}$ (MPa) | $m_i$ | D | $\gamma$ (kN/m$^3$) |
|---|---|---|---|---|---|
| Value Range | 15–40 | 25–35 | 15 | 0.7–0.9 | 25 |

The value of $\sigma_{ci}$ was taken by referencing Liao's study [25], in which $\sigma_{ci}$ ranges from 15 MPa to 40 MPa based on test data of sandstone collected from the Taliao Formation. According to in situ slope investigations, the bedrock was already subjected to long-term weathering. Therefore, in this study, the range of $\sigma_{ci}$ was appropriately reduced to 25 MPa to 35 MPa after omitting potentially low values that are deemed too conservative.

Based on in situ investigations of the slope bedrock performed by Sinotech Engineering Consultants, Inc., and comparisons with the GSI chart [26], the GSI of the hill slope is estimated to range from 20 to 40 (see Figure 11). Marinos and Hoek [27] estimated that heterogeneous rock mass (such as flysch) similar to the case of bedrock in this study has a GSI of 30. Marinos and Carter [28] noted that bedrock subjected to tectonism and weathering will have a significantly lower GSI, which was the case in this study. Therefore, the lower GSI limit in this study was set to 15 for conservative consideration.

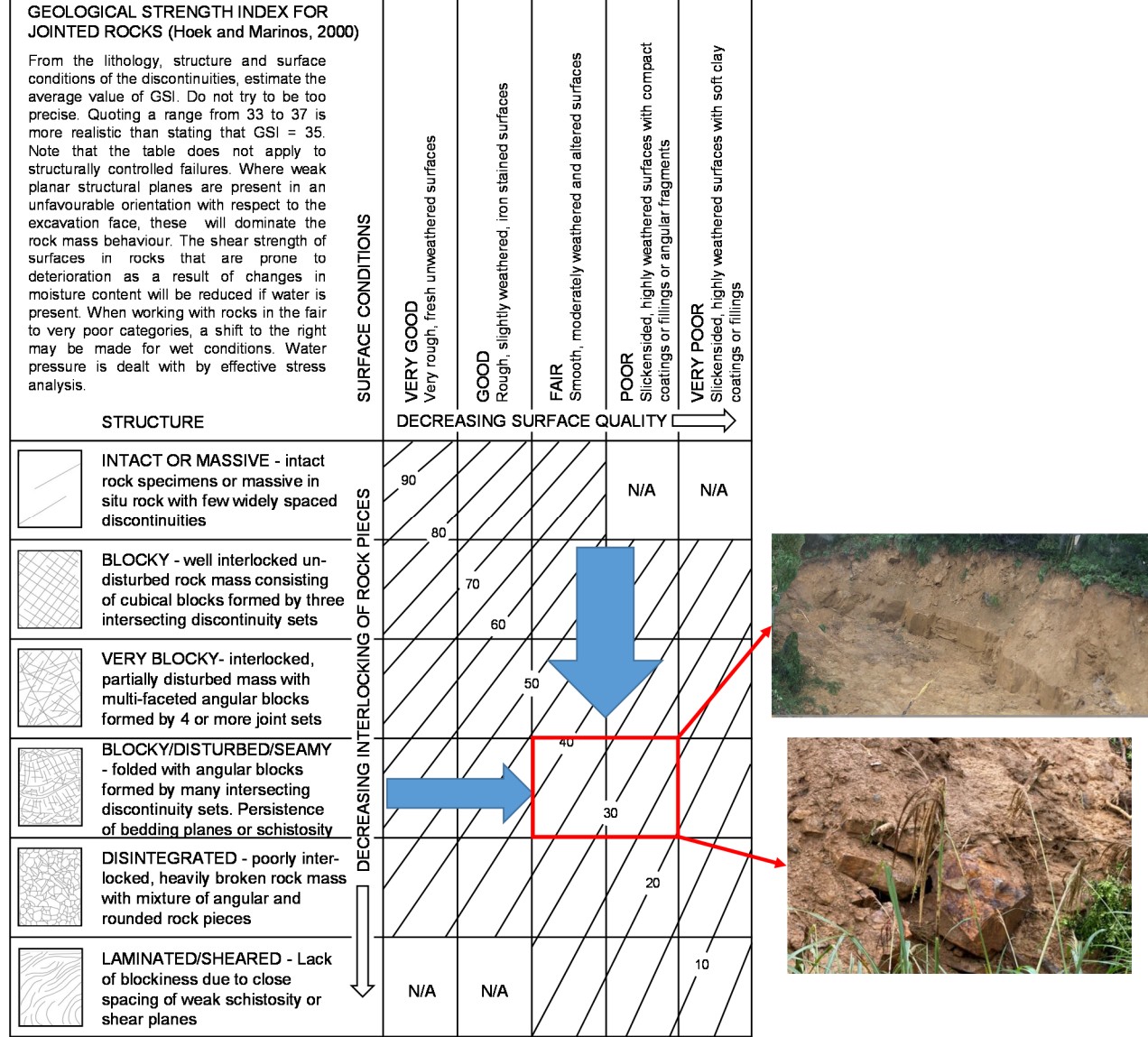

**Figure 11.** Classification of the studied rock slopes according to the basic GSI chart (adapted from Marinos and Hoek (2000) [26]).

In fact, to date, Taiwan lacks sufficient in situ assessment data regarding the GSI of slope bedrock. Therefore, the values obtained from slope bedrock grade assessments performed during portal excavation in tunnel engineering are used instead. Shao (1997) [29]

estimated that the rock mass rating (RMR) of slope bedrock during the excavation of the Xueshan Tunnel (formerly known as the Pinglin Tunnel) portal was around 10 to 25. Because the Xueshan Tunnel is also located at the northeastern region of Taiwan where the geology appears to be fractured, the landslide area is presumed to be similar. However, Hoek and Brown (1997) [30] previously mentioned that, in the case of poor bedrock conditions, a GSI should not be estimated directly using a low RMR if GSI $\leq$ 25. As mentioned previously, by using large-area rock mass strength mapping, the CGS classifies the bedrock in the study area as a Type IV close to poorer strength rock mass, which is rather close to the GSI estimated in this study with the influence of weathering taken into account. Consolidating the in situ investigations and large-area mapping data, the upper and lower slope GSI limits in the study area were set as 15 and 40, respectively. This range will serve as a reference for deducing the GSI of the bedrock slope.

To date, Taiwan does not have large-area in situ data on the D values of the slopes. Taiwan is situated on the Pacific Ring of Fire, where prolonged seismic forces have greater effects on the D value of bedrock in comparison with the small-range D values for tunnel excavation projects or slope blasting practices in mining. Under these circumstances, this study used a D value range of 0.7 to 0.9 as proposed by Hoek and Brown [17] for slope stability analysis. Afterwards, the appropriate D value for bedrock slopes in the northeastern region of Taiwan was deduced through the back analyses of the practical numerical simulation results.

The value of $m_i$ differs by rock mass type. Based on the suggestions of Marinos and Hoek (2001) [27], and considering the characteristics of sandstone at the Taliao Formation, the value of $m_i$ is initially set as $17 \pm 4$. Because similar values of $m_i$ had marginal effects on slope stability [31], $m_i$ is set as 15 in slope stability analyses. Given that the rock unit weight ($\gamma$) of sandstone is generally considered to range from 1.9 to 2.6 t/m$^3$, it is taken as 25 kN/m$^3$ in this study.

This study utilized the Optum$^{G2}$ slope analysis software that is grounded in the finite element-based upper and lower bound limit analysis methods. The software computes a gravity multiplier (GM) value based on the capacity of a material's elements to withstand external forces. The GM value represents the stability of a slope, and its significance is similar to the conventional FS of a slope. To evaluate the stability of a slope through the GM value, an input unit weight ($\gamma$) is multiplied with an aspect ratio and then the slope body beared stress is released until a near-failure critical state is achieved. The outcome is the unit weight that corresponds to a slope at a critical state ($\gamma_{cr}$). Afterwards, the GM value (aspect ratio either magnifying or reducing) that corresponds to the initial unit weight ($\gamma$) is calculated, and this GM value is the factor of safety of the slope (FS$_\gamma$). This concept is presented as Equation (8). For the sake of readability, FS$_\gamma$ is presented as the conventional FS in this study. Although Equation (8) is different from conventional FS via a limit equilibrium analysis, it can still reflect the mechanical significance of a slope stability analysis.

$$\text{FS}_\gamma = \frac{\gamma_{cr}}{\gamma} \tag{8}$$

## 4. Examination of Key Factors Pertaining to the Stability of Closely Jointed and Weathered Rock Mass Slopes

The Hoek–Brown failure criterion is suitable for rock slope stability analysis in this study because the case slope has a closely jointed and weathered rock mass structure [21]. Sonmez and Ulusay et al. (1998) [20] numerically simulated several real cases of rock slope failures and observed the presence of near-arc-shaped failures in rock mass that are closely jointed or have sufficiently fractured discontinuities. The reasonable parameters of the Hoek–Brown failure criterion can be deduced through back analyses of the numerical simulation results. A FELA-based numerical model was used in this study to simulate the process of the failure of closely jointed and weathered rock mass slopes in order to examine the effects of each key parameter of the Hoek–Brown failure criterion on the rock slope

stability and to determine the reasonable range for each key parameter that corresponds to similar bedrock slopes in the northeastern region of Taiwan.

*4.1. Comparison of the Numerical Simulation Results with Actual Rock Slope Failures*

This section discusses the types of slides at different stages of a landslide as well as the respective FSs.

According to the descriptions of the slope before the landslide occurred on November 30, it was determined that several days of continuous rainfall had increased the groundwater level, which resulted in the bulging of the temporary shotcrete constructed on the toe of the slope. Therefore, the toe had presumably sustained small-scale failures before the closely jointed and weathered rock mass slope failed completely. A further series of parametric investigations yielded several details that will be elaborated subsequently in this text, alongside the appropriate parameter value combinations for the failed slope. Figure 12 shows the FS and failure plane simulation results at different groundwater levels prior to the completely landslide. According to the results, even though there is an obvious failure plane at the toe of the slope for all three groundwater levels, Figure 12b,c indicate that the groundwater level does not reach the ground surface, and the respective FSs are 1.986 and 1.472. Compared to the FS of 0.675 at a high groundwater level (see Figure 12a), the likelihoods of a slide failure in both cases are significantly smaller. At a parameter value combination of $\sigma_{ci} = 30$ MPa, GSI = 20, and D = 0.9, and when the groundwater level is close to the ground surface, an obvious failure plane can be observed at the toe of the slope that is in line with the case landslide in this study. In other words, at a high groundwater level, a small-scale failure plane had occurred at the toe of the slope, and the FS of 0.675 corresponds to the Stage 1 process of the failure shown in Figure 13. On this basis, the groundwater level in Figure 12a is presumed to be closest to the actual landslide conditions, and at this moment, the initial stage of failure had occurred on the slope.

After the initial failure, TRA's monitoring personnel soon noticed a wider range of slope instability that ultimately triggered the landslide on the morning of December 4. Therefore, it is reasonable to assume that the internal structure of the slope rock mass gradually failed within this timeframe. Figure 13 shows the failure planes at different stages as the precipitation gradually increased the groundwater level (Stages 2 to 4 are briefly presented here). Even though the FS at Stage 3 (1.168) is slightly larger than that of Stage 1, according to Li et al. [31], who considered the variation of rock mass materials based on Hoek's suggestions [32], a value of 1.168 suggests a 40% likelihood of failure, and the slope remains at a near-failure critical state. At Stage 4, the FS had become smaller than 1 (0.964). Therefore, the simulation shows that after the toe of the slope was subjected to initial failure, the minimum principal stress of the slope mass behind the failure decreased due to lateral stress released. This equivalent to increasing maximum principal stress had increased the driving force on the top part of the slope, and the deviatoric stress acting on the rock mass exceeded the yield strength of the weathered rock mass, resulting in a retrogressive slope failure.

During the process of the retrogressive slope failure induced by the lateral stress released, the failure arc formed in each stage was removed from the numerical model (as indicated by the dotted lines in Figure 13), except for the one at the final stage during which the FS of the remaining slope mass at the top part of the slope was larger than 1 (see Figure 13, FS = 2.112 at the final stage). This indicates that the slope had completely slid off and attained its final stable state. The entire sliding process is similar to composite failure mode [33]. Comparing Figures 9b and 13 indicates that the simulated final stage was considerably closer to conditions during the actual landslide.

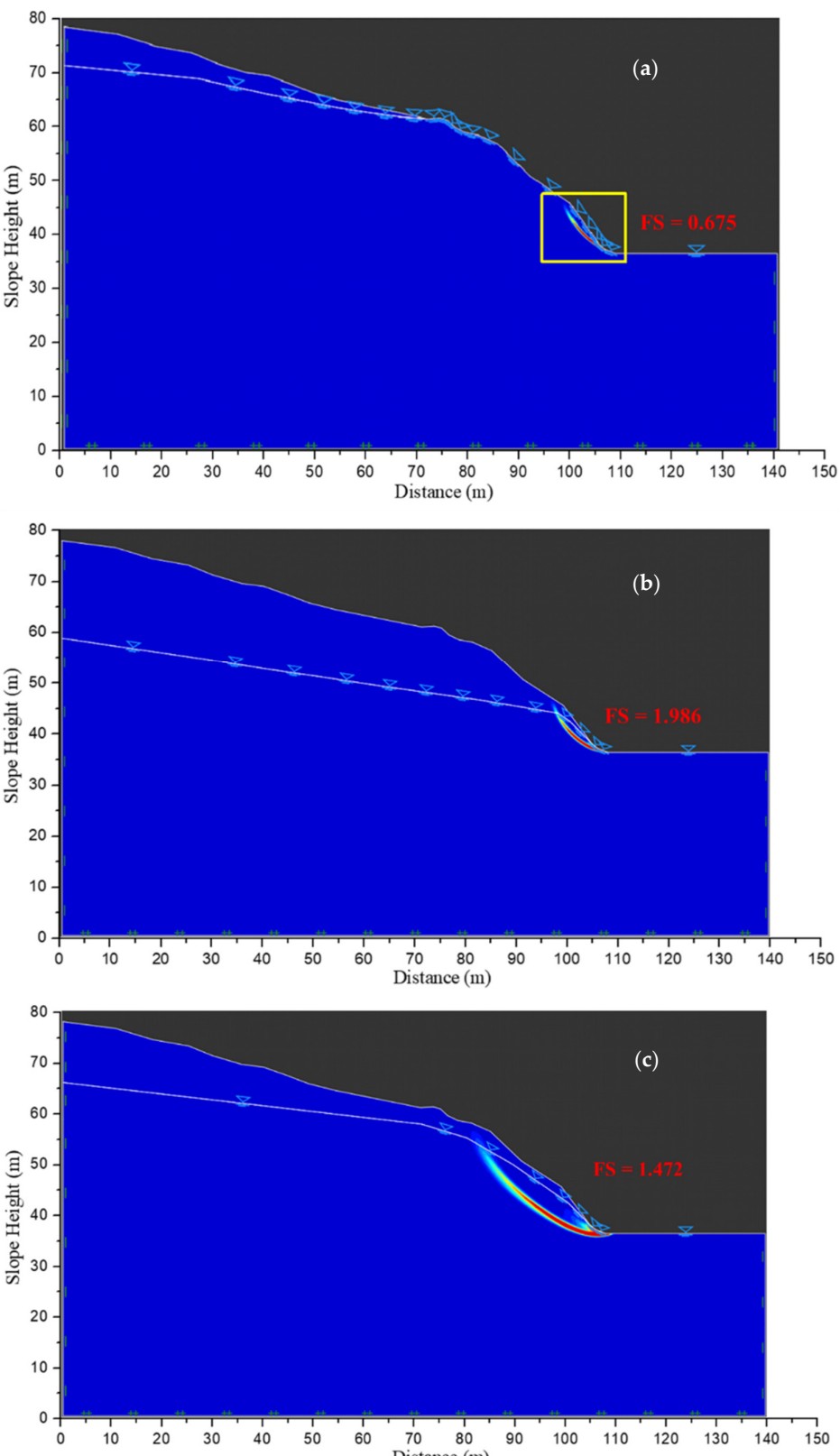

**Figure 12.** Initial failure at the toe of the slope at different groundwater levels (parameter value combination of $\sigma_{ci} = 30$ MPa, GSI = 20, and D = 0.9). (**a**) Groundwater level and failure plane conditions close to the case landslide. (**b**) Low groundwater level and failure plane conditions. (**c**) Moderate groundwater level and failure plane conditions.

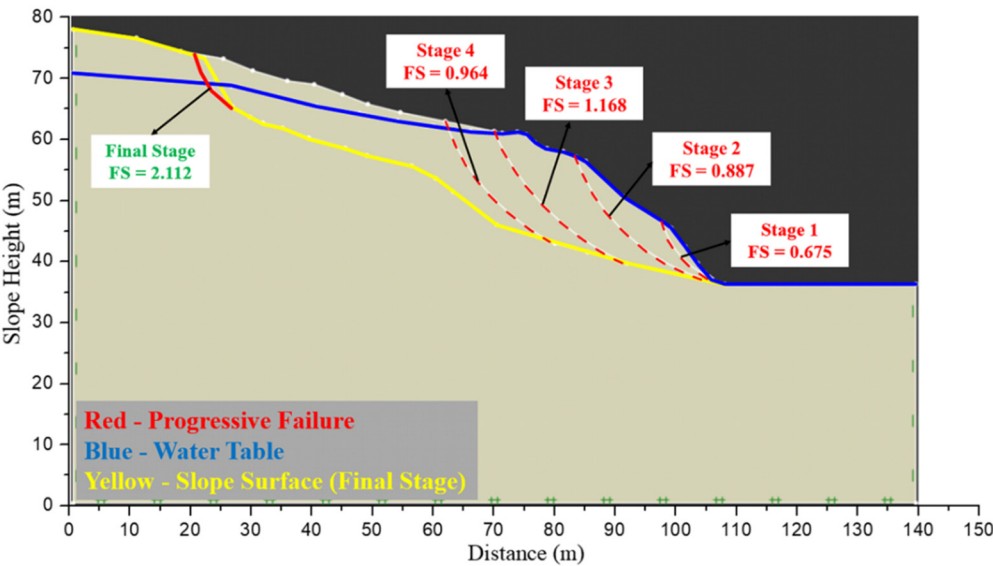

**Figure 13.** Simulation results of the different stages of retrogressive slope failure.

It is found that when the landslide occurred, the 24-h accumulated rainfall in the study area had merely attained the "heavy rain" standard (80 mm/24 h or 40 mm/h) per the CWB classification system, which differs considerably from the common perception in Taiwan that the likelihood of slope sliding is highest when the rainfall level must reach an "extremely heavy rain" standard (200 mm/24 h or 100 mm/3 h). Inferring from this, although the two weeks of continuous rainfall increased, groundwater level is one of the reasons caused this landslide. However, weathering on the closely jointed rock mass caused rock slope strength reduction is also the reason for sliding before reaching the extreme heavy rain level.

*4.2. Examination of the Key Parameters of the Hoek–Brown Failure Criterion in Rock Slope Stability Analysis*

Li et al. [31,34] analyzed rock mass using the Hoek–Brown failure criterion and showed that the key parameters of $\sigma_{ci}$, D value, and GSI had larger effects on slope stability analysis, whereas $m_i$ had smaller effects. According to the analysis results in the preceding section, the numerical simulation results of the model slope were in line with the actual process of the failure of the study slope. Therefore, the parameters used in the model were applied in subsequent examinations. While the D value and GSI are usually acquired via in situ investigations, due to the high level of difficulty of doing so at the site and the subjectivity of human judgments, this study collectively used the three parameters of $\sigma_{ci}$, D value, and GSI as a combination to examine their effects on closely jointed and weathered rock mass slope sliding, with the hopes that the reasonable ranges of the D value and GSI can be deduced via back analyses of a real case, thereby providing a reference for examining similar bedrock slopes. The numerical simulation process entailed multiple analyses of different value combinations of the parameters in Table 1. For the sake of brevity, only combinations of three $\sigma_{ci}$ values (27.5, 30, and 32.5 MPa), two D values (0.8 and 0.9), and three GSIs (15, 20, and 25) are presented in the text. Figure 14 presents the schematic of different parameter value combinations on the analyses. The following results describe the effects of different key parameters of the Hoek–Brown failure criterion on slope stability analysis.

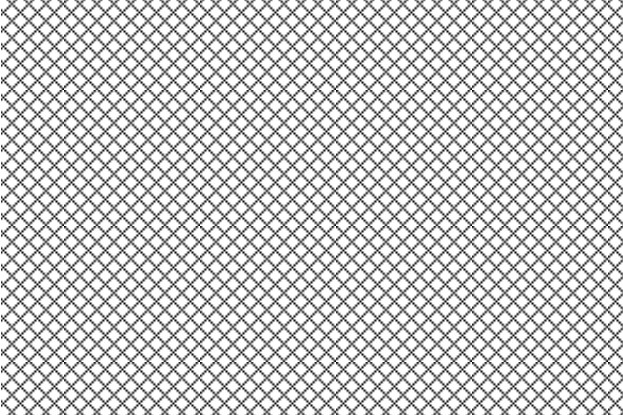

**Figure 14.** Schematic of the different combinations of $\sigma_{ci}$, D, and GSI used for analysis.

(i). As shown in Figure 15, at a fixed $\sigma_{ci}$ and GSI, rock mass with a smaller D value has a relatively larger FS at each stage. This suggests that rock mass subjected to higher degrees of natural disturbances would slide more easily. The significant differences between the D value and FS indicates that the D value has significant effects on the rock mass slope stability analysis. In practice, there is a lack of D values suitable for bedrock slopes in Taiwan. However, given that Taiwan is situated on a seismically active region, the authors suggest that the regional D value of bedrock in Taiwan is highly affected by seismic forces. Based on back analyses of the case landslide, when the D value ranges from 0.8 to 0.9, the simulation results are more in line with the actual conditions in the case landslide. Therefore, it is reasonably assumed that closely jointed and weathered sandstone rock mass slopes in the northeastern region of Taiwan have a D value ranging from 0.8 to 0.9. Additionally, the D value obtained from in situ slope investigations often falls within a certain range instead of a single value. Therefore, the D value of 0.8 to 0.9 derived in this study can be applied in slope stability analysis models of similar regions where the actual D value is difficult to obtain through in situ investigations.

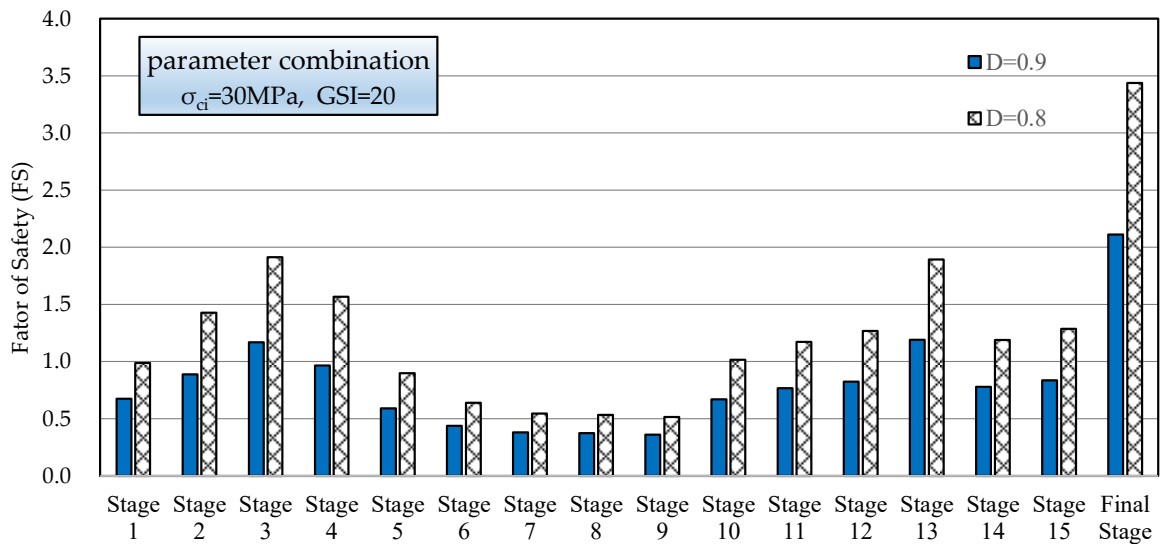

**Figure 15.** Effect of the D value on the FS in each stage (at a parameter value combination of $\sigma_{ci} = 30\,\text{MPa}, \text{GSI} = 20$).

(ii). As shown in Figure 16, at a fixed D value and GSI, the FS of rock mass with a large $\sigma_{ci}$ is greater than that of rock mass with a small $\sigma_{ci}$ in all stages. This suggests that an intact rock strength affects slope stability. However, the FSs of different $\sigma_{ci}$ vary marginally, which suggests that the effect of $\sigma_{ci}$ on rock slope stability is not as significant as the effect

of the D value. Hence, a rock slope stability analysis can be performed using representative and reasonably accurate values of $\sigma_{ci}$. As previously mentioned, even though the FSs of Stages 3 and 13 are slightly above 1, the likelihood of failure remains high in both stages.

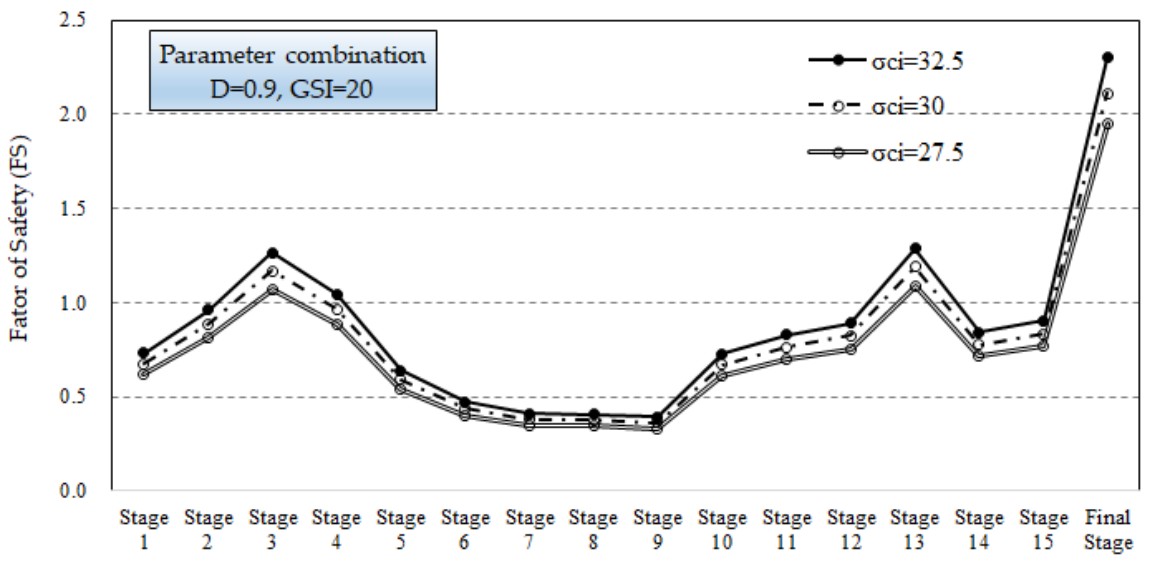

**Figure 16.** Effect of $\sigma_{ci}$ on the FS in each stage (at a parameter value combination of D = 0.9 and GSI = 20).

(iii). As shown in Figure 17, at a fixed $\sigma_{ci}$ and D value, the FS of rock mass with a large GSI is greater than that of rock mass with a small GSI in all stages. This suggests that bedrock slopes that are more intact have higher stability. Given that the FS varies substantially across different GSI values, it can be known that in comparison with $\sigma_{ci}$, the GSI of closely jointed and weathered rock mass has more significant effects on stability. Similar to the D value, the GSI is obtained through in situ slope investigations, and well-versed geotechnical engineer's assessing GSI would often be within a certain range and small deviation. Nonetheless, Figure 17 shows that the GSI is a sensitive parameter in slope stability analysis.

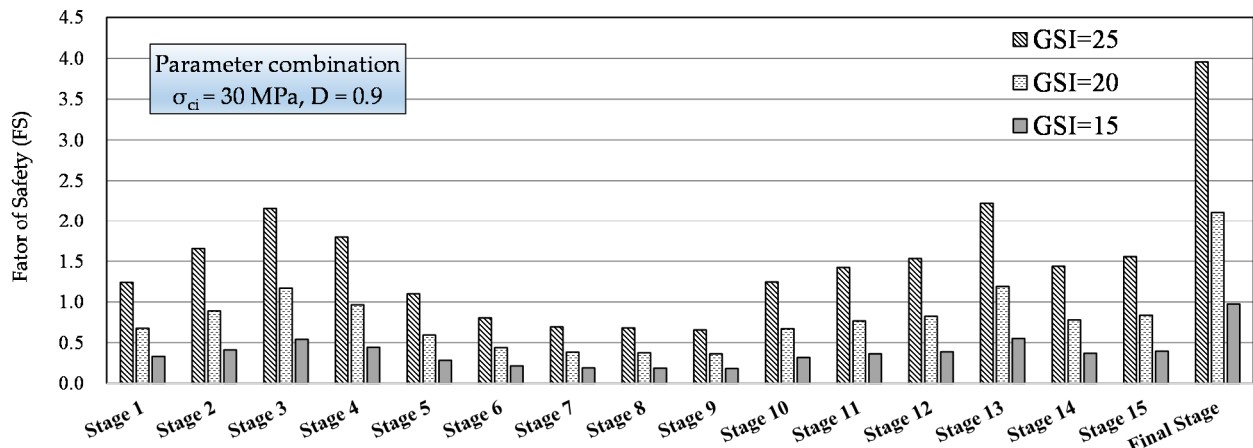

**Figure 17.** Effect of GSI on the FS in each stage (at a parameter value combination of $\sigma_{ci}$ = 30 MPa and D = 0.9).

(iv). The analysis results of the three aforementioned parameter value combinations indicate that the D value and the GSI of the Hoek–Brown failure criterion have more significant effects on slope stability analysis. Therefore, the reliability of the in situ values of these two parameters should be taken into consideration. In addition, D value, GSI, and $\sigma_{ci}$ can collectively affect the FS. When the FS is a single fixed value, any slight decrease in

highly sensitive D value or any slight increase in GSI will require a substantial reduction in the less sensitive $\sigma_{ci}$ in order to maintain a fixed FS. In this case, even though the $\sigma_{ci}$ range of 10 to 25 MPa presented in Figure 7c based on Franklin's system remains smaller than the range of 27.5 MPa to 32.5 MPa obtained through back analyses in this study, it is still considered a reasonable range. Based on this, in light of a lack of large-area in situ investigation data, consider that Taiwanese geotechnical engineers use the CGS's rock mass strength map as an indirection reference for selecting the sketchy rock mass parameters.

(v). The numerical simulation results suggest that even though the parameter value combinations of the Hoek–Brown failure criterion affect the FS in slope stability analysis, their effects on the position and range of the slip plane remain insignificant. This finding is in line with the study of Li et al. [31], and the reason for this is presumably because rock mass materials follow the nonlinearity of the Hoek–Brown failure criterion.

## 5. Conclusions

This study presents the case of a landslide that occurred on a slope that consists of closely jointed and weathered bedrock common in Taiwan. By means of finite element limit analysis and the Hoek–Brown failure criterion, this study successfully presented a process of slope failure similar to the actual landslide. In addition, this study computed the applicability of key Hoek–Brown failure criterion parameters for analyzing the stability of closely jointed and weathered sandstone slopes in the northeastern region of Taiwan. The results can assist in the engineering analysis of similar bedrock slopes in Taiwan. The findings of this study are as follows.

(i). The simulation process indicated that a small-scale failure had occurred first at the toe of the closely jointed and weathered rock mass slope due to an increased groundwater level caused by continuous rainfall. It agreed with in situ inspections performed on November 30 when the accumulated rainfall was 300 mm noted the presence of scattered rock-soil mixed debris fallen in the toe area. This situation represents the lateral stress released (a decreasing minimum principal stress) acting on the slope mass behind, resulting in a retrogressive slope failure. The landslide gradually occurred until attaining a final state, at a time when the accumulated rainfall had reached 600 mm. The initial to final stages of slope failure simulated through FELA were in line with actual conditions.

(ii). D value has significant effects on the stability of rock slopes. However, there is no reliable reference value for this parameter in Taiwan due to the lack of similar studies in the past. Because Taiwan is located in a seismically active region, the regional D value of bedrock in Taiwan is mainly affected by seismic forces. Back analyses of the parameters suggest that the suitable D value range for closely jointed and weathered sandstone rock mass slopes in the northeastern region of Taiwan is 0.8 to 0.9. Similar to the D value, the effects of GSI on slope stability are significant. Comparing the case landslide with the numerical simulation results, this study determined through back analyses that the suitable GSI range for the rock mass slope is 20 to 30. This range can serve as a reference for a wider area of applications in the future such as developing precautionary measures against, and assessments of, rock-slope hazards.

(iii). The Hoek–Brown failure criterion is suitable for analyzing the stability of closely jointed and weathered rock mass slopes. According to the analysis of the parameter value combinations, it is particularly important to take note of the significance and reliability of the D value and GSI in in situ applications. While the $\sigma_{ci}$ of rock mass affects slope stability, its effects are not as significant as those of the D value and GSI. Therefore, slope stability analysis can be performed using several representative values of $\sigma_{ci}$, while the significance and reliability of the D value and GSI must be accounted for in in situ investigations.

(iv). Taiwan has many similar slopes where the bedrock is weathered, closely jointed and with a high groundwater level. More landslide cases could be added to develop a Hoek–Brown failure criterion parameter database for rock mass slopes in Taiwan. Furthermore, not only Hoek–Brown system but also other rock slope evaluating method for example

Q-slope classification system could be considered, thereby enriching the multiplicity or enhancing the applicability on broad-area rock slope preliminary stability analysis.

**Author Contributions:** K.-S.S. and A.-J.L. proposed the idea and designed the research; K.-S.S., A.-J.L. and C.-N.C. completed the discussions; K.-S.S. and A.-J.L. performed numerical analyses; K.-S.S. and C.-H.C. deal with data curation; C.-F.L. and C.-P.K. investigated and collected field data; K.-S.S. and A.-J.L. wrote original paper. All authors have read and agreed to the published version of the manuscript.

**Funding:** The authors would like to thank the Taiwan Building Technology Center from The Featured Areas Research Center Program within the framework of the Higher Education Sprout Project by the Ministry of Education in Taiwan for their support.

**Institutional Review Board Statement:** Not applicable.

**Informed Consent Statement:** Not applicable.

**Data Availability Statement:** The data that support the findings of this study are available from the corresponding author upon reasonable request.

**Acknowledgments:** The authors also gratefully acknowledge (i) Central Weather Bureau of Taiwan (CWB) for providing the rainfall data, and (ii) National Science and Technology Center for Disaster Reduction (NCDR) for providing and free downloaded rainfall distribution picture.

**Conflicts of Interest:** The authors declare no conflict of interest.

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
