# Peer review of "Investigations of a Weathered and Closely Jointed Rock Slope Failure Using Back Analyses"

_sustainability, doi:10.3390/su132313452_

Round 1

Reviewer 1 Report

1. The article compares the numerical simulation result with the actual landslide. If the theoretical calculation can be combined, the structure of the article will be more perfect
2. The article uses a certain amount of space to introduce FELA and Hoek-Brown failure criterion. Because these contents are familiar to everyone, this part of the content can be simplified appropriately.
3. Some pictures in the text are not beautiful enough. As part of the paper, it is recommended to adjust the picture format appropriately to meet the specifications of scientific papers.
4. In recent years, there have been many research results on the influence of rainfall on landslides. It is recommended that the author further strengthen the collation and reading of the literature.

Reviewer 2 Report

Review of the manuscript titled "Investigations of a weathered and closely jointed rock slope failure using back analyses"

The manuscript presents an interesting case study using Hoek-Brown failure criterion in conjunction with FELA. The work is interesting, but the quality of illustrations are very poor. Also, the conceptual part should be completely rewritten, as most space is dedicated to well known concepts. I suggest the authors to revise the manuscript thoroughly to consider the same for publication.

Reviewer 3 Report

Title: Investigations of a weathered and closely jointed rock slope failure using back analyses

Journal / ID: Sustainability / sustainability-1464376

Authors: Shao Kuo-Shih, Li, An-Jui, Chen Chee-Nan, Chung Chen-Hsien, Lee Ching-Fan, and Kuo Chih-Ping

Dear Gentlemen/Madam,

Thank you very much for giving me an opportunity for revising the manuscript entitled “Investigations of a weathered and closely jointed rock slope failure using back analyses” by Shao Kuo-Shih and his/her colleagues were submitted to the Sustainability. The manuscript is founded on the application of empirical, numerical and remote-sensing-based assessment for landside instability in the northeastern region of Taiwan. The manuscript used GSI classification, FEM modeling and GIS-based susceptibility mapping for demonstrating the studied landslide condition. Mainly work is contains the regular task of a field report regardless of significant scientific achievements. The authors have to pay more attention to the scientific contribution to their work. Nevertheless, the following comments can be useful to modify the current situation of the manuscript:

Comment 1: Considering the existing literature, the motivation of the manuscript is not convincing. There is a large literature associated with numerical modeling, empirical classification systems, and GIS-based susceptibility assessment for landslides. Various techniques have been applied/developed to assess the modeling better. I expect to see a novelty in a manuscript dealing with comparative stability assessment. I do not think that new case studies would help the scientific community to go one step further. The result of this manuscript is only valid for the examined area. Based on site-specific conditions and the quality of a modeling, different results can be obtained in another site, and another technique may appear like the best alternative. However, in the end, these efforts do not help us decide on the best method of assessments. So, it will appreciate if the authors provide a solid contribution and highlight the novelty of the manuscript (as per my observation).

Comment 2: The title is not relevant to the manuscript contents. Landslides can provide composite movements on large scale (as mentioned in the text), but jointed rock slopes are usually considered specific failures. Please check this paper: https://www.tandfonline.com/doi/full/10.1080/17538947.2021.1988163.

Comment 3: The concluding remarks of the abstract are not well-written. It’s merely the repetition of the objectives in the manuscript. The abstract contains the aims of the study, describes the case, methodology, finding and achievements, advance of method, and conclusions. Please rewrite the abstract as well as possible.

Comment 4: It will be appreciated if the authors ordered the manuscript as Introduction, analysis method, studied case, method and materials, results and discussion, verification and justifications, conclusion. Each section has to provide relevant information.

Comment 5: The necessity of the manuscript should be presented in the introduction section. The appropriate introduction contains a process framework on underground excavations, coal mining, rock mass instabilities, numerical modeling, literature of related work and the necessity of your work that justified your work is superior to others.

Comment 6: What type of landslide was analyzed?; Needs to describe correctly. Please consider this paper: https://www.mdpi.com/2220-9964/10/5/315.

Comment 7: Control the references and use the proper format to demonstrate documentation. There are many “Error! Reference source not found” in the text.

Comment 8: The methodology section is weakly written. So, my suggestion is to reconstruct it. For example, use a process flowchart. The basics of the FEM procedures are well-known. There is no need to present the basics, the scientific articles are not a book or chapter of students. The authors have to stress what they do with FEM to reach their goal. Also, it can also stand for empirical and remote sensing procedures as well.

Comment 9: Using general references like “Facebook” or “Wikipedia” is not proper for scientific papers. The authors have to understand that they providing scientific research. So, it will appreciate if the authors strongly consider the tip.

Comment 10: The manuscript is not clear that the authors what to want to do?. Are they want to provide the rock slope stability or landslide susceptibility assessment?. There is a big difference between these tasks. Please consider these papers:

https://www.sciencedirect.com/science/article/abs/pii/S0013795221002969; https://www.sciencedirect.com/science/article/abs/pii/S0266352X20303694; https://www.sciencedirect.com/science/article/pii/S2405844020307520; https://www.sciencedirect.com/science/article/abs/pii/S0266352X19303040; https://www.sciencedirect.com/science/article/abs/pii/S0341816220303830; https://www.sciencedirect.com/science/article/abs/pii/S0341816219305685; https://www.sciencedirect.com/science/article/abs/pii/S0013795219312347; https://www.koreascience.or.kr/article/JAKO202111236745328.page;

Comment 11: In my opinion, authors should better explain findings in the results and discussion section instead of the conclusion section. Also, the authors should deepen the discussion.

Comment 12: The application of empirical relations is a very good move to classify the rock slope mass. But providing a single method is not appropriate to cover the task. So, it will appreciate if the authors provide comparative empirical classifications for the studied slope like GIS, SMR, and Qslope. Please consider these papers:

https://link.springer.com/article/10.1007/s00603-017-1305-0; https://link.springer.com/article/10.1007/s10706-021-01991-w; https://www.koreascience.or.kr/article/JAKO201931961768643.j; https://link.springer.com/article/10.1007/s10064-019-01459-5; https://www.sciencedirect.com/science/article/abs/pii/S0013795219306726; https://link.springer.com/article/10.1007/s00603-020-02176-2;

Comment 13: Please add a subsection clearly articulating the main limitations, wider applicability of your methods, and findings in the discussion section.

Comment 14: I noticed that the conclusion section tends to repeat abstract and results. The conclusion paragraph should be short, impactful, and direct the reader to this research’s next steps and opportunities.

Comment 15: The English of the manuscript is readable; however, I would suggest the authors check it again to avoid any mistakes.

Best regards,

Round 2

Reviewer 1 Report

accept

Author Response

Thanks for your valuable comments.

Reviewer 2 Report

The authors have attempted addressing the reviewers concerns. I suggest the authors to add the responses and changes made in the manuscript to the response note, for better understanding.

I suggest one more round of revisions for the manuscript, and some minor comments are mentioned in the attached file. 

Reviewer 3 Report

The manuscript generally improved. it can be move forward to next stage.

Author Response

Thanks for your valuable comments.